# A PCR-RFLP Technique to Assess the Geographic Origin of *Plasmodium falciparum* Strains in Central America

**DOI:** 10.3390/tropicalmed7080149

**Published:** 2022-07-26

**Authors:** Gustavo Fontecha, Denis Escobar, Bryan Ortiz, Alejandra Pinto

**Affiliations:** Microbiology Research Institute, Universidad Nacional Autónoma de Honduras, Tegucigalpa 11101, Honduras; denis.escobar@unah.edu.hn (D.E.); bryan.ortiz@unah.edu.hn (B.O.); mpinto@unah.edu.hn (A.P.)

**Keywords:** malaria, Central America, *pfs47*, *pfs48/45*, *pvs47*, Honduras, *Plasmodium falciparum*

## Abstract

The elimination of malaria requires strengthening diagnosis and offering adequate and timely treatment. Imported cases of falciparum malaria represent a major challenge for pre-elimination areas, such as Central America, where chloroquine and primaquine continue to be used as first-line treatment. The *pfs47* gene has been previously described as a precise molecular marker to track the geographic origin of the parasite. The aim of this study was to design a simple and low-cost technique using the polymorphic region of *pfs47* to assess the geographic origin of *P. falciparum* strains. A PCR-RFLP technique was developed and evaluated using the *MseI* enzyme that proved capable of discriminating, with reasonable precision, the geographical origin of the parasites. This method could be used by national surveillance laboratories and malaria elimination programs in countries such as Honduras and Nicaragua in cases of malaria where an origin outside the Central American isthmus is suspected.

## 1. Introduction

The incidence of malaria in the Americas has decreased by 67% between 2000 and 2020. Three South American countries (Venezuela, Brazil and Colombia) account for more than three quarters of the cases on the continent [1]. The seven countries of the Central American isthmus show an unequal epidemiological scenario. While El Salvador and Belize no longer report indigenous cases, in Nicaragua malaria has overflowed to levels similar to those of 20 years ago, with 31,763 confirmed cases in 2020, which represents 88% of the entire subregion.

Historically, *Plasmodium falciparum* malaria in Honduras has accounted for 10 to 20% of all cases [2]; however in 2020 it exceeded 29% and reached more than 43% of cases in 2021 (personal communication with the National Malaria Surveillance Laboratory, Health Ministry, Honduras). According to the World Malaria Report 2021, cases considered imported by Central American countries range from 0.08% in Nicaragua to 24.1% in Costa Rica. The number of imported cases into Honduras comes mainly from Nicaragua and has been increasing, from 0.06% in 2014 to 15.6% in 2019 [1]. There are few documented reports of imported malaria cases from outside the region. Until epidemiological week 23/2022, Honduras registered 18 imported cases from Nicaragua and one case originating in Uganda (personal communication with the Surveillance Health Unit, Health Ministry, Honduras). In 2019, a case of malaria imported from Kenya in a pediatric patient was reported in Honduras [3]. In a genomic surveillance study at the Hospital Escuela in the city of Tegucigalpa, two cases of falciparum malaria resistant to chloroquine imported from Africa were found [4].

Given the increasing mobility of people through Central America to the United States [5], the introduction of *Plasmodium* sp. strains from South America or the Old World represents an additional challenge for the public health systems of countries with limited resources such as Honduras. The introduction of virulent *P. falciparum* strains that require a treatment scheme different from that indicated by the National Standard with chloroquine and primaquine [6] is a risk for the management of imported cases.

Due to the interest in monitoring imported cases in countries or regions where malaria has been eliminated, some molecular methods have been described as an alternative to decipher the probable geographic origin of the parasites. Some methods analyze panels of single nucleotide polymorphisms (SNPs) by amplification and sequencing of *P. vivax* and *P. falciparum* genomes [7,8,9,10]. In China, a panel of 26 microsatellites has been used to assess the geographic origin of imported cases of *P. falciparum* [11]. These methods can discriminate, with acceptable precision, the geographical origin of the parasites; however, they require facilities with sequencing capabilities, highly qualified personnel, and financial resources, which limits operational implementation at the country level and in cross-border areas. In 2021, Molina-Cruz et al. developed a simpler and more efficient technique to establish the geographic origin and transmission potential of malaria cases of unknown origin using PCR-based high-resolution melting (HRM) [12]. This assay is based on the genotyping of two highly informative SNPs within domain 2 of the *P. falciparum pfs47* gene.

PfS47 belongs to the family of proteins of six cysteines (6-Cys) [13] along with PfS230 and PfS48/45, which are promising candidates as transmission-blocking vaccines (TBV) [14,15]. This family of proteins is expressed on the surface of gametes and is important for the fertility of *P. falciparum* [16]. In addition to its role in sexual reproduction, *pfs47* is essential for the evasion of the mosquito’s immune response by inhibiting JNK-mediated apoptosis by preventing caspases activation [17] and suppressing midgut nitration responses critical for activating the mosquito complement-like system [18]. To adapt to the wide diversity of anopheline species present in the tropics of four continents, *pfs47* has been subjected to natural selection that has led to a strong population structure [19,20]. This adaptation has resulted in the fixation of a few polymorphisms within the second domain of the gene that divides the parasite into haplotypes compatible with the anopheline species present on each continent and that follow a “lock and key” logic [21].

In this study, we propose a simple and low-cost PCR-RFLP assay to assess the geographic origin of *P. falciparum* strains based on three SNPs within *pfs47*. This method could be used in the national surveillance laboratories of Central America to evaluate the cases of imported malaria that occur in each country in the current scenario that seeks elimination by 2030.

## 2. Materials and Methods

### 2.1. In Silico Analysis of the pfs47 Gene and Search for Candidate Restriction Enzymes

To analyze the variability in sequence polymorphisms within the *pfs47* gene, the NCBI database was queried through the nucleotide BLAST tool. Three hundred sixty-five available sequences of the gene were downloaded and imported into the Geneious software^®^ 9.1.7 (Biomatters Ltd., Auckland, New Zealand). Sequences were trimmed to delimit a segment of 94 base pairs that included positions 707 and 725, demonstrated as polymorphic by Molina et al. [12]. A multiple alignment of all sequences was performed to identify sequence polymorphisms. The country of origin of each sequence was recorded. Subsequently, an in silico search for candidate restriction enzymes capable of cutting exactly at the SNPs of interest was performed.

### 2.2. Construction of Phylogenetic Trees Based on Nucleotide and Amino Acid Sequences of pfs47

To demonstrate the correlation between polymorphisms within domain 2 of *pfs47* and the geographic origin of the parasite, a new search for *pfs47* sequences longer than 1282 bp was performed in GenBank. Fifty-six sequences with a size of at least 1280 bp were downloaded. These 56 sequences were chosen because they were representative of four continents: 18 sequences from African countries, 23 sequences from Asia and Oceania, and 15 sequences from the Americas. Sequences were aligned using the Global Alignment method with free end gaps in the Geneious software with a cost matrix of 51% similarity, and a gap open penalty of 3. The alignment was used to build a phylogenetic tree using the UPGMA method and Tamura Nei as a model of genetic distance and bootstrap of 500 replicates. Sequences were translated using the correct open reading frame (ORF) to produce putative sequences 427 amino acids long. These peptides were aligned, and a second phylogenetic tree was generated. Subsequently, the 56 nucleotide sequences of *pfs47* were trimmed down to 94 bp in the region within domain 2 described as polymorphic [12]. The alignment of these sequences was used to construct a third phylogenetic tree. Finally, the shortened 94 bp sequences were translated to generate putative peptides of 31 residues whose alignment was used to construct a fourth phylogenetic tree.

### 2.3. Amplification and Sequencing of the pfs47, pfs48/45, and pvs47 Genes

A 94 bp segment encompassing the polymorphic region of domain 2 of the *pfs47* gene was amplified. DNA was extracted from 32 blood samples deposited on filter paper and previously diagnosed as cases of falciparum malaria from Honduras and Nicaragua [6].

For DNA extraction, three 10 mm^2^ circles were cut from blood impregnated Whatman filter paper. Samples were immersed in 200 µL of 1% saponin and incubated at 4 °C overnight. The next day, samples were washed four times with phosphate-buffered saline (PBS), then resuspended in Chelex-100 (Bio-Rad, Hercules, CA, USA), incubated at 56 °C for 15 min and at 100 °C for 10 min. The tubes were centrifuged for 5 min at 13,000 rpm and the DNA present in the supernatant was recovered and stored at 4 °C for later analysis. In addition, DNA was extracted from *P. falciparum* isolate 04/176, used as an international DNA standard for nucleic acid amplification assays [22], as well as from the reference strain K1, native to Thailand, the Dd2 strain from Southeast Asia, the Brazilian strain 7G8, and the HB3 strain from Honduras. A strain of *P. falciparum* isolated in Honduras from a merchant seaman infected in West Africa was also tested. The DNA was extracted using the AutoMate™ system and the PrepFiler Express Forensic DNA Extraction Kit™ (Applied Biosystems, Waltham, MA, USA) according to the manufacturer’s instructions. PCR reactions were performed in a volume of 50 μL, with 25 μL of 2× master mix Taq polymerase (Promega Corp. Madison, WI, USA), 3 μL of each primer 10 μM (Table 1), 14 μL of nuclease-free water, and 5 μL of DNA (40 ng/μL). The reactions were carried out by an initial denaturation at 95 °C for 5 min, 40 cycles of 95 °C for 30 s, the annealing temperature of 52 °C for 30 s, and 72 °C for 30 s, with a final extension at 72 °C for 5 min. Amplicons were visualized by 3% agarose gel electrophoresis with ethidium bromide. Positive and negative controls were included in each set of reactions. The amplicons were sequenced from both sides at the Psomagen facility (Rockville, MD, USA), and the sequences were analyzed in the Geneious^®^ 9.1.7 software searching for polymorphisms at positions 707 and 725 of the gene.

A partial segment of approximately 567 bp of the *pfs48/45* gene was amplified by semi-nested PCR. Fifteen samples of *P. falciparum* isolates from Honduras and Nicaragua and strain 04/176 were amplified. Both reactions were performed in a volume of 50 μL, with 25 μL of 2× master mix Taq polymerase (Promega Corp. Madison, WI, USA), 2 μL of each primer 10 μM (Table 1), 16 μL of nuclease-free water, and 5 μL of DNA (40 ng/μL). The PCR product of the first reaction was used as a template for the second PCR. The amplification program for both reactions consisted of an initial denaturation at 95 °C for 1 min, 35 cycles of 95 °C for 1 min, 54 °C for 1 min, and 72 °C for 1 min, with a final extension at 72 °C for 5 min. The partial sequences of 502 nucleotides obtained in this study were compared with 36 homologous sequences downloaded from GenBank reported from at least three countries from each of the following three continents: Africa, Asia, and America. Sequences were aligned to assess the number of polymorphic sites.

A 1032 bp fragment of the *P. vivax pvs47* gene was also amplified from three samples diagnosed with vivax malaria in Honduras in a previous study [23]. The amplification reaction included 25 μL of 2× master mix Taq polymerase, 1 μL of each primer 10 μM (Table 1), 16 μL of nuclease-free water, 2 μL of 20 mg/mL bovine serum albumin, and 5 μL of DNA (40 ng/μL). The amplification program consisted of an initial cycle of 95 °C for 5 min, 37 cycles of 95 °C for 1.5 min, 60 °C for 1.5 min, and 72 °C for 2 min, with a final extension at 72 °C for 10 min. Amplicons of *pfs48/45* and *pvs47* were visualized by 1% agarose gel electrophoresis with ethidium bromide. The amplicons were sequenced from both sides and analyzed using the Geneious^®^ 9.1.7 software. The partial sequences of 1034 nucleotides obtained in this study were compared with 70 homologous sequences downloaded from GenBank reported from two countries in the Americas and five Asian countries. Sequences were aligned to calculate the number of polymorphic sites.

The sequences obtained for *pfs48/45* and *pvs47* were deposited in the NCBI GenBank and accession numbers were assigned.

### 2.4. PCR-RFLP of pfs47

The amplification products of the 94 bp segment were digested with the enzyme *Tru1I* (*MseI*) (cutting site T^TAA) (Thermo Fisher Scientific Inc., Waltham, MA, USA) in 31 μL as the final volume including 18 μL of nuclease-free water, 2 μL of 10× buffer B, 10 μL of PCR product, and 1 μL of the enzyme 10 U/μL. The reaction was incubated at 65 °C for 8 h. Undigested products and digested fragments were separated on a 12% nondenaturing polyacrylamide gel (PAGE) at 20 v and 5 mAmp and Gel Loading Dye, Orange (6×). Subsequently, the gels were stained with ethidium bromide in 1× TBE buffer for 15 to 30 min.

## 3. Results

### 3.1. In Silico Analysis of the pfs47 Gene and PCR-RFLP Using Tru1I (MseI) to Reveal Informative Polymorphic Sites

A total of 365 *pfs47* sequences with a length of approximately 1319 bp were downloaded from GenBank. The sequences were shortened to 94 bp in the region including positions 661 to 754 of the partial cds of the accession number KT892054.1. Six polymorphisms were found at positions 671, 707, 718, 725, 740, and 742 (Figure 1). An in silico analysis was performed testing 5640 restriction enzymes available in the Geneious^®^ 9.1.7 software, which could cut at the two positions of interest, 707 and 725. Only the enzyme *Tru1I* (*MseI*) (T^TAA) was able to cut simultaneously at both positions. Of 365 sequences, 96 showed the T707/C725 haplotype, 38 showed the T707/T725 haplotype, and the remaining 231 sequences showed the C707/C725 haplotype. In the entire 94 bp fragment, *MseI* was able to cut between one and four times at the following positions: 700^701, 707^708, 724^725, and 742^743. Therefore, the putative restriction pattern consisted of two to five bands (Figure 2).

Four restriction patterns were identified (Figure 2) among the 56 sequences analyzed. Pattern A had two bands (54/40 bp) derived from the CCA haplotype at positions 707/725/742. Pattern B revealed three bands (42/40/12 bp) with CCT haplotype. Pattern C showed four bands (40/35/12/7 bp) with the TCT haplotype, and pattern D showed five bands (40/18/17/12/7 bp) with the TTT haplotype. Patterns A and B were more common on the African continent, although representatives of all four restriction profiles were found. Pattern C was the most common in Asia, although one strain with pattern D (from Papua New Guinea) and another strain with pattern B were also found. Pattern D was the most common in Latin America, but a Colombian isolate with pattern C and a Brazilian isolate with pattern B were found.

### 3.2. Phylogenetic Trees Based on Nucleotide and Amino Acid Sequences of pfs47

Fifty-six *pfs47* sequences greater than 1282 nucleotides were downloaded and translated, and putative peptides were aligned to detect the most polymorphic regions within the gene. As shown in Figure 3, the *pfs47* gene is composed of three highly conserved domains rich in cysteine residues. Domain 2 has a variable region that harbors at least five mutations at sites 236, 240, 242, 247, and 248 of the protein. Two possible residues, threonine or isoleucine are observed at position 236 of the protein. Positions 240 and 248 are occupied by leucine or isoleucine; 242 may be occupied by serine or leucine, and 247 may be occupied by valine or alanine. In positions 236 and 242 of the polypeptide, which correspond to polymorphisms 707 and 725 of the gene, mutations change a neutral polar amino acid for a neutral non-polar one or vice versa. At positions 240, 247, and 248, the residues were nonpolar neutral.

To demonstrate whether there was a correlation between mutations within the variable region of domain 2 of *pfs47* and the geographic origin of the parasites, four phylogenetic trees were constructed based on the alignment of both nucleotide and putative peptide sequences. Trees were constructed using the nearly complete nucleotide sequence of the gene (Figure 4a), the 94 bp variable sequence of the second domain (Figure 4b), the 427-residue peptide sequence (Figure 4c), and a partial sequence of 31 residues from domain 2 (Figure 4d). In the four cladograms, a separation of clusters was observed based on the geographic origin of the parasites. 5 to 6 strains (10–11%) were not within the expected geographic clade.

### 3.3. Sequencing of the pfs47, pfs48/45, and pvs47 Genes

Thirty-one 94-bp fragments of *pfs47* isolated from Honduras and Nicaragua were sequenced and the T707/T725 haplotype was demonstrated in all of them. This involves isoleucine at position 236 of the polypeptide and leucine at 242, forming an I236/L242 haplotype. The reference strain 04/176 showed the haplotype C707/C725, with a haplotype T236/S242. The sequences were not deposited in GenBank because they were too short. In addition, a partial segment of approximately 567 bp of the *pfs48/4*5 gene was sequenced in 15 samples from Honduras and Nicaragua. No polymorphisms between sequences were observed. The BLAST tool showed 100% similarity to the *P. falciparum* HB3 strain with accession number EF137242.1. Comparing the *pfs48/45* sequences obtained here with 36 sequences from nine countries on three continents, 482 of 502 non-variable sites (96%) with pairwise % identity equal to 99.5% were observed. A sequence was deposited in GenBank with accession number ON715433. Finally, a 1036 nucleotide fragment of the *pvs47* gene was sequenced from three samples from patients with vivax malaria in Honduras. The sequences also did not show polymorphisms and one of them was deposited in GenBank under accession number ON715434. The number of polymorphic sites of *pvs47* relative to 70 homologous sequences from seven other countries was 1011 of 1034 (97.8%), and the pairwise % identity was 99.7%.

### 3.4. PCR-RFLP of pfs47

To test whether the predicted in silico restriction analysis was transferable to practice, the 94-bp polymorphic segment of *pfs47* was amplified from 32 Central American strains and some reference strains isolated from Honduras, South America, Southeast Asia, and Africa. The PCR products were digested with the enzyme *MseI* and separated on a non-denaturing polyacrylamide gel (Figure 5). Three restriction patterns were obtained: Pattern A (54/40 bp) was found in strain 04/176 and in the West African blood sample. Pattern C (40/35 bp) was observed in the K1 strain from Thailand and the Dd2 strain from Southeast Asia. A pattern like the “D” pattern predicted by in silico analysis (40/18 bp) was observed in the 7G8 strain from Brazil, HB3 from Honduras, and in all samples from Honduras and Nicaragua. In pattern D, a slight inconsistency was detected between the observed lower band (>25 bp) and the expected one (18 bp). No sample presented the pattern B predicted in silico. Bands less than 30 bp in size could not be seen on the gel.

## 4. Discussion

Imported cases of malaria are a challenge for non-endemic countries and for regions certified free of malaria transmission. In particular, cross-border transmission from highly endemic countries to malaria-free regions is a continuing threat to national surveillance programs [26]. Asian countries like Kuwait [27], Malaysia [28], Turkey [29], and China [30,31], among others, report a high number of malaria cases from neighboring countries or Africa. North African countries, with no indigenous cases, also report a significant number of imported malaria cases from the sub-Saharan region [32]. In countries like Senegal, with a highly stratified transmission, from low in the dry north to moderately high in the humid south [33], there is evidence of imported cases between regions [34].

In the Americas, the epidemiology of malaria is uneven, meaning that non-endemic countries such as Chile, which receives many migrants from South America and Africa, report dozens of imported cases each year [35]. Currently, social, economic, and political situations favor human migration between South American countries. This is the case of gold miners moving between Venezuela, Guyana, and northern Brazil, generating a disproportionate number of cases of imported malaria [1,36]. The mass migration caused by the humanitarian and political crisis in Venezuela in recent years, with more than 3 million people forced to leave their country, has resulted in an increase in malaria cases in neighboring countries, mainly Colombia [37].

In Central America, published reports of imported malaria are rare, and most are anecdotal. According to the World Malaria Report 2021, cases officially reported in 2020 as imported by Central American countries ranged from 0.08% in Nicaragua to 24.1% in Costa Rica. In 2020, Honduras reported 98 imported cases out of 913 (10.7%), mainly from Nicaragua [1]; however, these data come only from interviews conducted as part of passive surveillance activities. Using a whole-genome selective amplification approach of 59 *P. vivax* samples from Panama, two imported cases from Brazil, one from India, and one from China were identified [38]. A clinical case of pediatric malaria imported from Kenya was reported in 2019 in Honduras [3], and a recent genomic surveillance study detected two cases of falciparum malaria imported from Africa with chloroquine resistance haplotypes [4]. Despite the limited availability of imported malaria reports, the growing number of migrants from Central America, Haiti, South America, and other low-income countries in Africa and Asia traveling across the isthmus to reach the United States is overwhelming and raises concern about the potential introduction of more virulent strains of *P. falciparum* [5].

The identification of the origin of malaria cases is usually done through interviews with patients to collect demographic data and travel history [36,39]; however, it is not always possible to trace the exact place where patients became infected. A few methods have been developed to accurately trace the geographic origin, particularly of *P. falciparum* isolates. A technique based on the amplification of mitochondria and apicoplast genomes reveals the region of origin of *P. falciparum* isolates by means of a 23-SNP barcode based on single-step PCR and sequencing [7,8]. A similar approach has been proposed for *P. vivax* by characterizing a barcode based on SNPs with high predictive power [9]. Barcoding has also been used to assign the geographic origin of cases imported into the United States [10]. Another approach for geographic origin assignment of imported falciparum malaria cases in China applied a panel of 26 microsatellite markers with acceptable accuracy results [11]. More recently, a High-Resolution Melting (HRM)-based technique to genotype two SNPs in the *pfs47* gene of *P. falciparum* proved to be simple, reproducible, and accurate enough to assign geographic origin at the continent level (Americas, Africa, and Asia/Oceania) [12].

All these methodologies are innovative and can be implemented in laboratories in countries that have the necessary technology and highly qualified human resources. In this study, we develop a simpler and cheaper PCR-RFLP technique based on the polymorphisms of domain 2 of the *pfs47* gene described by Molina-Cruz [12]. This technique is based on the presence or absence of four restriction targets within a 94 bp fragment of the *pfs47* gene, which allows parasite isolates to be assigned at the continental level with reasonable precision. Restriction patterns A and B are more frequent in Africa, and pattern C is more common in Asia, while pattern D predominates in Latin America. Although genotype D does not seem to be exclusive to the Americas, all strains isolated in Honduras and Nicaragua, as well as the reference strain HB3 from Honduras, exclusively showed pattern D. Therefore, the finding of any of the other genotypes in a patient with suspected imported malaria in Honduras or Nicaragua would suggest a geographic origin outside of the Central American region.

Pattern D observed in the experiments with strains from Central America showed a band greater than 40 bp, as predicted in silico; however, the expected band of lower molecular weight (18 bp) presented an anomalous migration pattern above the 25 bp. This anomalous slow mobility has been described for certain restriction fragments on polyacrylamide gels, especially for DNA molecules containing 4–6 contiguous adenine residues (Figure 1) that might have curved helical backbones and behaving electrophoretically as if they were larger than their actual sizes [40]. In any case, despite this migratory phenomenon in electrophoresis, pattern D is sufficiently diverse from the other patterns to distinguish strains of Central American origin.

The approach proposed in this study has the limitation that genotype D would not guarantee with absolute certainty that it is not a case imported from South America, Africa, or even Papua New Guinea, and it would be necessary to apply other molecular markers to resolve the situation. A marker that could be used for this purpose is *pfcrt*, since strains of *P. falciparum* from Honduras, Guatemala, and Nicaragua show a wild-type genotype sensitive to chloroquine, very rarely found in other regions [6]. One of the main limitations of this study is the low number of local samples analyzed, but taking into account the drastic decrease in malaria cases in Honduras and the low genetic diversity demonstrated in parasite populations [41], the probability that most or all of the circulating strains are of genotype D is high. Likewise, due to its simplicity and low cost, the PCR-RFLP technique can be implemented in national epidemiological surveillance laboratories that require a minimum infrastructure for molecular biology such as thermocycler and vertical electrophoresis equipment, and that do not have a real-time PCR machine with HRM analysis capacity, or a DNA sequencing machine.

On the other hand, the phylogenetic trees obtained here confirm what has been described by several authors in relation to the strong geographic structure observed in the *pfs47* gene [12,19,20,21], which can be particularly attributed to polymorphisms within domain 2, as demonstrated in Figure 4. In this study, partial sequences of the *pfs48/45* and *pvs47* genes were also obtained from local strains of parasites. These genes dramatically affect the fertility of parasites [24] and have been proposed as promising candidate molecules for malaria transmission-blocking vaccines (TBV) [42,43]. The low variability found in both genes coincides with the low genetic diversity of circulating parasites in Central America [41] and with what is described in the literature [25,42,44]. These data are a contribution that will help in the development of TVB’s potentially effective against malaria parasites regardless of geographic region.

## 5. Conclusions

Here we present an inexpensive, simple, and rapid technique based on the genotyping of *pfs47* SNPs with the aim of evaluating the geographic origin of *P. falciparum* isolates causing malaria imported from outside the Central American isthmus. This method could be used in national malaria epidemiological surveillance laboratories, particularly in the current scenario that seeks to eliminate malaria by 2030.

## Figures and Tables

**Figure 1 tropicalmed-07-00149-f001:**
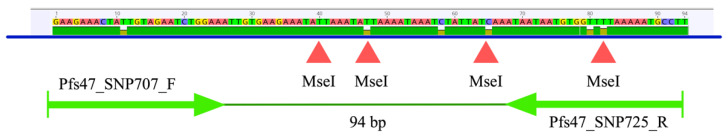
94 bp fragment of the *pfs47* gene indicating the primer target sites and four *MseI* cutting points.

**Figure 2 tropicalmed-07-00149-f002:**
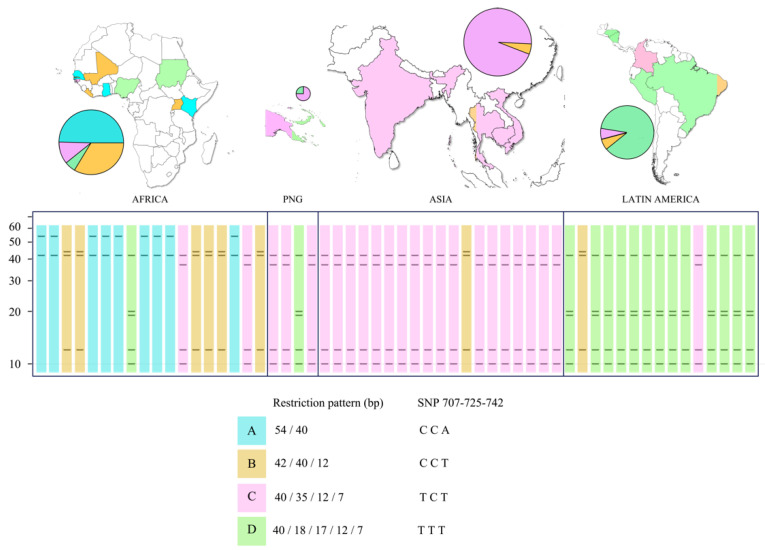
Restriction profiles of the *pfs47* gene digested with *MseI* in relation to the geographic origin of the isolates.

**Figure 3 tropicalmed-07-00149-f003:**
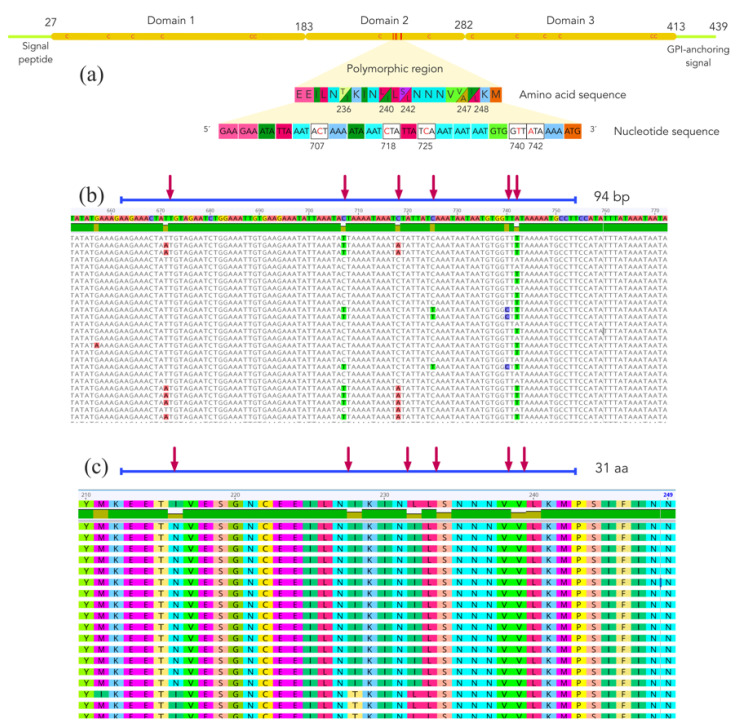
(**a**) Structure of the *Plasmodium falciparum* cysteine-rich protein PfS47, showing three domains and a variable region within domain 2. The letter “C” within the polypeptide chain indicates the approximate position of the cysteine residues. (**b**) Multiple nucleotide sequence alignment of the polymorphic region of the *pfs47* gene. (**c**) Multiple amino acid sequence alignment of the polymorphic region of the PfS47 protein. Red arrows indicate polymorphic sites.

**Figure 4 tropicalmed-07-00149-f004:**
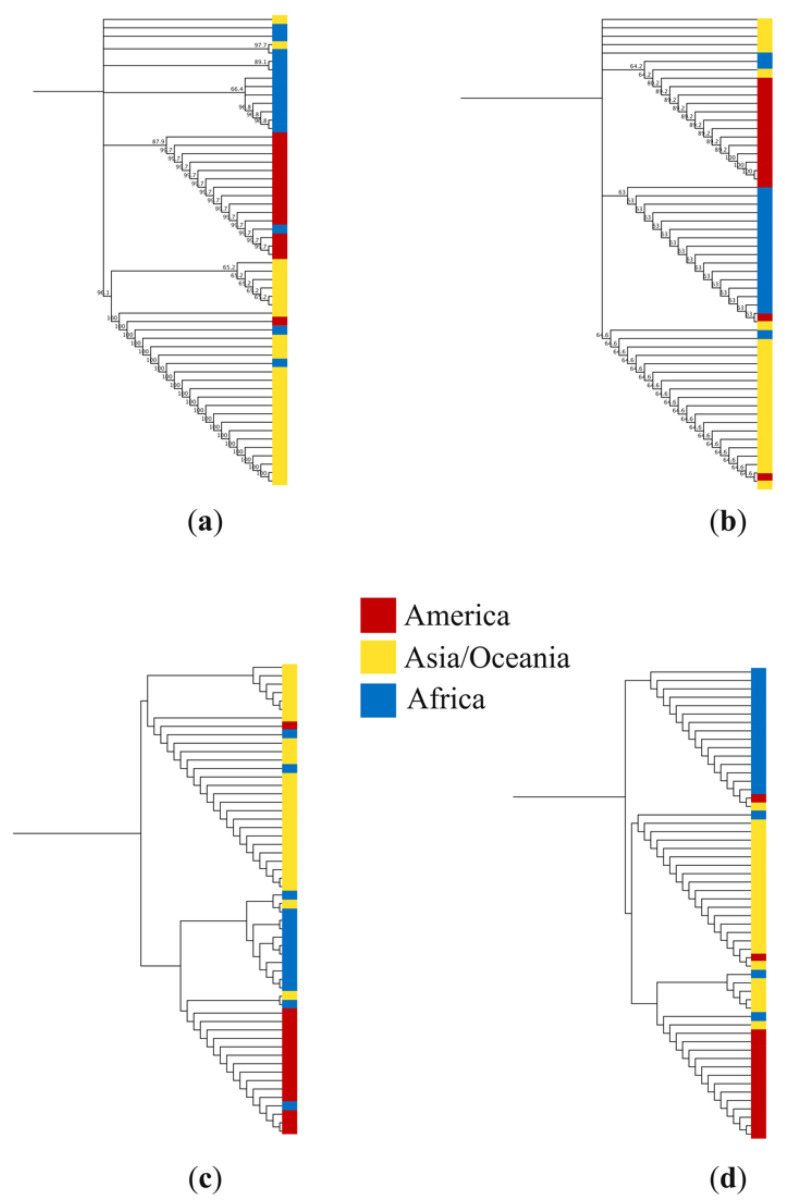
Cladograms constructed with the UPGMA method using (**a**) 56 sequences of 1282 nucleotides of the *pfs47* gene; (**b**) sequences of 94 nucleotides from domain 2 of the *pfs47* gene; (**c**) sequences of 427 residues of the putative protein PfS47; and (**d**) 56 sequences of 31 predicted residues of domain 2 of the PfS47 protein. Colors indicate the geographic origin of *Plasmodium falciparum* isolates.

**Figure 5 tropicalmed-07-00149-f005:**
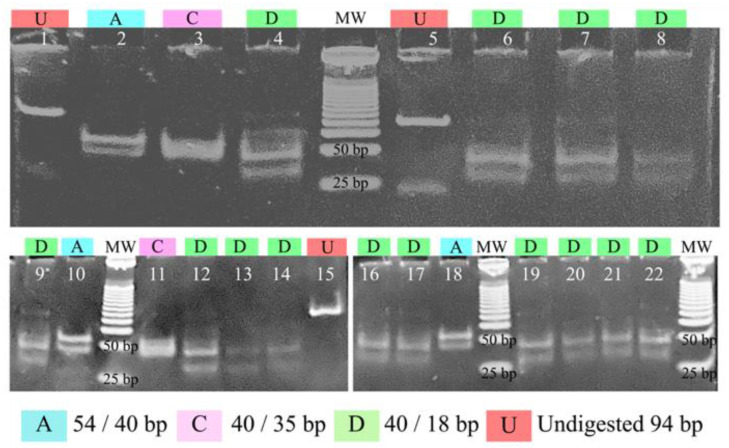
Polyacrylamide gel electrophoresis shows the restriction patterns of a fragment of the *pfs47* gene digested with the enzyme *MseI*. Lanes 1, 5, and 15 show undigested amplification products, lanes 2 and 10 show strain 04/176, lane 18 shows a parasite of African origin isolated in Honduras, lane 3: strain K1, lanes 4, 6–9, 12–14, 16–17, 19–20: strains from Honduras and Nicaragua, lane 11: strain Dd2, lane 21: strain 7G8, and lane 22: strain HB3. MW: molecular weight marker of 25 bp.

**Table 1 tropicalmed-07-00149-t001:** Primers used to amplify the *pfs47*, *pfs48/45*, and *pvs47* genes.

Primer Name	5′-3′Sequence	Gene	Product (bp)	References
Pfs47_SNP707_F	GAAGAAACTATTGTAGAATCTGGAAA	*pfs47*	94	Molina Cruz et al. 2021 [12]
Pfs47SNP725_R	AAGGCATTTTTATAACCACATTATTA			
Pfs48/45_F1	GATCTTTTTACATATTTGCCG	*pfs48/45 1st round*	577	Anthony et al. 2007 [24]
Pfs48/45_R	CTTCATAATATTCAATATCTCC			
Pfs48/45_F2	GATCTTTTTACATATTTGCCG	*pfs48/45 2nd round*	532	Anthony et al. 2007 [24]
Pfs48/45_R	CTTCATAATATTCAATATCTCC			
Pvs47_F	CACACCACCGCAAACAGG	*pvs47*	1525	Woo et al. 2013 [25]
Pvs47_R	GTGCACATTCCGCGGTTG			

## Data Availability

Data supporting the conclusions of this article are included within the article and its additional files. The raw data used and/or analyzed during the present study are available from the corresponding author on reasonable request.

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
