# Peer review of "A PCR-RFLP Technique to Assess the Geographic Origin of *Plasmodium falciparum* Strains in Central America"

_tropicalmed, 2022, doi:10.3390/tropicalmed7080149_

Round 1

Reviewer 1 Report

Dear authors,

This manuscript is well written and designed in overall. 

Please check reference 1, Organization, WH. is not how WHO should be displayed in Zotero/end note.

However :

The gel pictures in fig5. are worrying me about the lack of migration time. Molecular markers bands are very close to each others, and according to fig2. there are missing some bands. Could you clarify the disperencies between expected results in fig2. and the gel pictures fig 5. ? If possible new pictures of fig 5. with better resolution, horizontal migration and better migration would be a good upgrade.

Author Response

Reviewer #1:

  1. Please check reference 1, Organization, WH. is not how WHO should be displayed in Zotero/end note.

A/ Thanks. We have corrected reference 1.

  1. The gel pictures in fig5. are worrying me about the lack of migration time. Molecular markers bands are very close to each others, and according to fig2. there are missing some bands. Could you clarify the disperencies between expected results in fig2. and the gel pictures fig 5? If possible new pictures of fig 5. with better resolution, horizontal migration and better migration would be a good upgrade.

A/ The reviewer is correct. There is an apparent contradiction in the restriction pattern "D". The answer to this apparent discrepancy is that the in silico prediction shows 4 bands (40, 18, 17, and 12 bp). In the real experiment, bands 18 and 17 are not resolved due to limitations of the technique and are observed as a single band. Bands less than 15 bp are not visible on 12% acrylamide gels and therefore are not taken into account in the restriction profile. In addition, the 18 bp band seems to migrate less than it should, but our hypothesis is that it is an intrinsic phenomenon to the migration of low molecular weight bands. In any case, the most relevant thing is that the 3 restriction patterns (A, C D) are clearly distinguishable which allows identifying the geographical origin of the isolates. It has not been possible to obtain a better image to replace the current one.

Reviewer 2 Report

Recommendation: Publish after major revisions noted.  

Comments:  

This manuscript describes a simple and low-cost PCR-RFLP assay to assess the geographic origin of P. falciparum strains based on three SNPs within pfs47. The authors need to address the following comments and revise the manuscript accordingly. 

  1. Please address the current transmission dynamics of P. falciparum. 
  2. Page 2, line 77: Please consider to add a diagram for in silico analysis and the workflow. 
  3. Page 3, line 103: Briefly describe the genomic DNA extraction process. 
  4. Highlight relative infectivity of P. falciparum clinical isolates.
  5. Discuss distribution of Pfs47 haplotypes in Central America.
  6. Highlight molecular form identification, infection status, mosquito characteristics and diversity indexes.

Author Response

Reviewer #2

This manuscript describes a simple and low-cost PCR-RFLP assay to assess the geographic origin of P. falciparum strains based on three SNPs within pfs47. The authors need to address the following comments and revise the manuscript accordingly. 

1. Please address the current transmission dynamics of P. falciparum.
Answer. The following paragraph has been included in the introduction: “Historically, Plasmodium falciparum malaria in Honduras has accounted for 10 to 20% of all cases [2]; however in 2020 it exceeded 29% [3] and reached more than 43% of cases in 2021(Personal communication by the National Malaria Surveillance Laboratory, Health Ministry, Honduras)”.

2. Page 2, line 77: Please consider to add a diagram for in silico analysis and the workflow.
Answer.The authors consider that a diagram representing the in silico analysis and a workflow diagram are unnecessary. The manuscript has a high number of diagrams of the in silico analysis (figures 1, 2, and 3), and the workflow is too simple (DNA extraction, PCR, REFLP and electrophoresis) to require an extra diagram.

3. Page 3, line 103: Briefly describe the genomic DNA extraction process.
Answer. DNA extraction has been described as follows: “For DNA extraction, three 10 mm2 circles were cut from blood impregnated Whatman filter paper. Samples were immersed in 200 µL of 1% saponin and incubated at 4 °C overnight. The next day, samples were washed four times with phosphate buffered saline (PBS), then resuspended in Chelex-100 (Bio-Rad), incubated at 56 °C for 15 minutes and at 100°C for 10 minutes. The tubes were centrifuged for 5 minutes at 13,000 rpm and the DNA present in the supernatant was recovered and stored at 4°C for later analysis”.

4. Highlight relative infectivity of P. falciparum clinical isolates.
Answer. We request that the reviewer be more explicit and let us know how this is relevant to the goals of the manuscript.

5. Discuss distribution of Pfs47 haplotypes in Central America.
Answer.The information requested by the reviewer is exactly the reason for this manuscript. The distribution of pfs47 genotypes in Central America is not known. The data provided by our results suggest that the strains from the north of the isthmus are homogeneous and that they could mostly show the “D” genotype, as indicated in the following paragraph: “One of the main limitations of this study is the low number of local samples analyzed, but taking into account the drastic decrease in malaria cases in Honduras and the low genetic diversity demonstrated in parasite populations [41], the probability that most or all of the circulating strains are of genotype D is high”.

6. Highlight molecular form identification, infection status, mosquito characteristics and diversity indexes.
Answer.We really fail to understand the relevance of what the reviewer is asking for. Perhaps this information (eg vector populations) will be relevant in future work studying the P47 receptor. In any case, our group has already published several works in relation to all these topics and that are tangential to the current manuscript.

Reviewer 3 Report

This article is well-written and highlights the spread of virulent strains of malaria with migration. The malaria parasite acquires various mutations that lead to the development of resistance against approved antimalarials. In this article, the author described an inexpensive, simple, and rapid technique based on the genotyping of pfs47 SNPs. This method could evaluate the geographic origin of P. falciparum isolates causing malaria imported from outside the Central American countries. This method could be helpful and provide a baseline for national malaria epidemiological surveillance laboratories. 

I have the following minor comments on this study.

  • Line no 183 "Pattern C was the most common in Asia, although one strain with pattern D and 183 another strain with pattern B were also found". 
  • I could not correlate this line with figure 2, as I could not see pattern D in the Asia region. Please modify the sentence.
  • In figure 4 the cladogram looks similar and, hard to see the pattern. It would be great if you could increase the figure size. 
  • In Line 99 it was mentioned about the MSA of 94 bp. Could you please provide the MSA (Nuceiotide and amino acid) only for this region of Pfs47 to give more clarity about the SNPs?

- Authors described P.vivax malaria. Please cite the following papers in lines no 48 and 294 as they are the pioneer in P. vivax diagnosis and sequencing of the genome.  

  • Kochar DK, Saxena V, Singh N, Kochar SK, Kumar SV, Das A. Plasmodium vivax malaria. Emerg Infect Dis. 2005 Jan;11(1):132-4. doi: 10.3201/eid1101.040519. PMID: 15705338; PMCID: PMC3294370.
  • Saxena, V., Garg, S., Tripathi, J., Sharma, S., Pakalapati, D., Subudhi, A. K., Boopathi, P. A., Saggu, G. S., Kochar, D. K., Kochar, S. K. and Das, A., Ashis Das. Plasmodium vivax apicoplast genome: A comparative analysis of major genes from Indian field isolates. Acta Tropica 122 (2012) 138-149.

In the reference section, a few scientific names need to be italicized. Please see attached file. 

Author Response

Reviewer #3:

  1. I have the following minor comments on this study. Line no 183 "Pattern C was the most common in Asia, although one strain with pattern D and 183 another strain with pattern B were also found". I could not correlate this line with figure 2, as I could not see pattern D in the Asia region. Please modify the sentence.

A/ The reason for the apparent discrepancy is that the strain expressing pattern "D" was isolated in Papua New Guinea. For the purposes of this study, PNG was included along with strains from the Asian region. However, the paragraph has been expanded as follows for clarity: “Pattern C was the most common in Asia, although one strain with pattern D (from Papua New Guinea) and another strain with pattern B were also found.”

  1. In figure 4 the cladogram looks similar and, hard to see the pattern. It would be great if you could increase the figure size.

A/ The reviewer is right. Figure 4 has been sent to the journal in high resolution, and it will be up to the designer to give it the most appropriate size so that it is understandable. The size of the figures is beyond the control of the authors of this work.

  1. In Line 99 it was mentioned about the MSA of 94 bp. Could you please provide the MSA (Nuceiotide and amino acid) only for this region of Pfs47 to give more clarity about the SNPs?

A/ A new version of Figure 3 has been provided that includes the two alignments suggested by the reviewer. The legend of the new figure is: (a) Structure of the Plasmodium falciparum cysteine-rich protein PfS47, showing three domains and a variable region within domain 2. The letter "C" within the polypeptide chain indicates the approximate position of the cysteine residues. (b) Multiple nucleotide sequence alignment of the polymorphic region of the pfs47 gene. (c) Multiple amino acid sequence alignment of the polymorphic region of the PfS47 protein. Red arrows indicate polymorphic sites.

  1. Authors described P.vivax malaria. Please cite the following papers in lines no 48 and 294 as they are the pioneer in P. vivax diagnosis and sequencing of the genome.

Kochar DK, Saxena V, Singh N, Kochar SK, Kumar SV, Das A. Plasmodium vivax malaria. Emerg Infect Dis. 2005 Jan;11(1):132-4. doi: 10.3201/eid1101.040519. PMID: 15705338; PMCID: PMC3294370.

Saxena, V., Garg, S., Tripathi, J., Sharma, S., Pakalapati, D., Subudhi, A. K., Boopathi, P. A., Saggu, G. S., Kochar, D. K., Kochar, S. K. and Das, A., Ashis Das. Plasmodium vivax apicoplast genome: A comparative analysis of major genes from Indian field isolates. Acta Tropica 122 (2012) 138-149.

A/ Although the two references suggested by the reviewer are pioneering and highly relevant, they are not helpful in supporting the specific ideas of lines 48 and 294, that is, citing methods to decipher the geographical origin of the parasite.

  1. In the reference section, a few scientific names need to be italicized. Please see attached file.

A/ Scientific names have been italicized in references.

Reviewer 4 Report

The authors have developed a relatively simple and accurate PCR-RFLP method for determining the geographic origin of Plasmodium falciparum, the causative agent of malaria. This method relies on the amplification of a 94bp polymorphic region of the Pfs 47 gene followed by digestion with the restriction with MSe1. Separation of the ethidium bromide fragments on non-denaturing  polyacrylamide gels and visualization reveals four distinct patterns : A, B,C and Ad. Patterns A and B are most commonly found in Africa ( where C and D also occur); pattern C and D respectively are found in Asia and Latin America. The method proposed by the authors is less tedious that similar methods requiring whole genome sequencing and heavy duty computing.

Despite the simplicity  of the method, its sensitivity and specificity were not determined in this study. Mixed infections with P.vivax, P malariae are common in some regions and this could be a confounding factor.At the same time no blinded samples were tested to validate the accuracy of the method. The authors should discuss these issues and nuance their conclusions accordingly. The paper is well written with only minor stylistic slips eg lines  89 and 331 with sentences that begin with figures.

In conclusion I recommend the article for acceptance subject to satisfactorily addressing the issues raised above.

Author Response

Reviewer #4:

  1. Despite the simplicity of the method, its sensitivity and specificity were not determined in this study. Mixed infections with P.vivax, P malariae are common in some regions and this could be a confounding factor.

A/ It is right. Indices such as sensitivity or specificity were not calculated in this study because it was not the objective of our work. To calculate these values, it would be necessary to analyze a large number of samples and an “n” with statistical robustness. This would mean having many strains from different geographical origins, however, in our laboratory we only have strains from the north of Central America and some reference strains. All of them were analyzed and reported here. In relation to mixed infections caused by non-falciparum species, this is not a problem for the test because the primers are specific for falciparum. Added to this, in Honduras and Nicaragua there are only Pv and Pf infections.

  1. At the same time no blinded samples were tested to validate the accuracy of the method. The authors should discuss these issues and nuance their conclusions accordingly.

A/ In fact, all the analyzed samples can be considered as blinded since the technique had not been described before and we could not know in advance the result of the digestion.

  1. The paper is well written with only minor stylistic slips eg lines  89 and 331 with sentences that begin with figures.

A/ Thanks. In both cases the numbers have been replaced by words.

Reviewer 5 Report

Fontecha et al. developed a simple RFLP technique to trace the origins of P. falciparum by targeting the polymorphic region of the Pfs47 gene fragment. An interesting piece of methodology, which can be adapted to use at the point of care laboratories to trace the country of origins of malaria species. The manuscript is well written and the authors provided detailed methodology. The manuscript is suitable for publication, however before accepting the manuscript, the authors need to clarify the following points.

Major concern: As per the in silico analysis for Pfs47, the RFLP pattern D produces the 40/18/17/12 bp fragment after digestion with MseI (Fig 2). However, fig 5 shows a completely different RFLP pattern and the authors are claiming that is pattern D, likely for the central American origin strain. This will mislead the data representation or the authors need to provide the full gel picture for figure 5.

Author Response

Reviewer #5:

  1. The manuscript is suitable for publication, however before accepting the manuscript, the authors need to clarify the following points. Major concern: As per the in silico analysis for Pfs47, the RFLP pattern D produces the 40/18/17/12 bp fragment after digestion with MseI (Fig 2). However, fig 5 shows a completely different RFLP pattern and the authors are claiming that is pattern D, likely for the central American origin strain. This will mislead the data representation or the authors need to provide the full gel picture for figure 5.

A/ The reviewer is correct. There is an apparent contradiction in the restriction pattern "D". The answer to this apparent discrepancy is that the in silico prediction shows 4 bands (40, 18, 17, and 12 bp), but bands 18 and 17 are not far enough apart in the actual experiment, and bands less than 15 bp are not visible on acrylamide gels. In addition, the 18 bp band seems to migrate less than it should, but our hypothesis is that it is an intrinsic phenomenon to the migration of low molecular weight bands. In any case, the most relevant thing is that the 3 restriction patterns (A, C D) are clearly distinguishable which allows identifying the geographical origin of the isolates.

Round 2

Reviewer 1 Report

Thanks for the clarification

Author Response

Thanks to you